# Developmental and Reproductive Impacts of Four Bisphenols in *Daphnia magna*

**DOI:** 10.3390/ijms232314561

**Published:** 2022-11-23

**Authors:** Lingling Qian, Chen Chen, Liguo Guo, Junping Deng, Xiangling Zhang, Jiexiang Zheng, Genmei Wang, Xiaofei Zhang

**Affiliations:** 1Innovation Center for Sustainable Forestry in Southen China, College of Forestry, Nanjing Forestry University, Nanjing 210037, China; 2Key Laboratory of Pesticide Environmental Assessment and Pollution Control, Nanjing Institute of Environmental Sciences, Ministry of Ecology and Environment, Nanjing 210042, China

**Keywords:** bisphenol analogs, developmental toxicity, reproductive toxicity, behavioral traits

## Abstract

Bisphenol A (BPA) is a typical endocrine-disrupting chemical (EDC) used worldwide. Considering its adverse effects, BPA has been banned or strictly restricted in some nations, and many analogs have been introduced to the market. In this study, we selected three representative substitutes, BPS, BPF, and BPAF, along with BPA, to assess the developmental and reproductive effects on *Daphnia magna*. The F0 generation was exposed to bisphenols (BPs) at an environmentally relevant concentration (100 μg/L) for 21 d; then the embryo spawn at day 21 was collected. Behavior traits, the activity of antioxidant enzymes, and gene transcription were evaluated at three developmental stages (days 7, 14, and 21). Notably, body length, heart rate, and thoracic limb beating were significantly decreased, and *D. magna* behaved more sluggishly in the exposed group. Moreover, exposure to BPs significantly increased the antioxidant enzymatic activities, which indicated that BPs activated the antioxidant defense system. Additionally, gene expression indicated intergenerational effects in larvae, particularly in the BPAF group. In conclusion, BPA analogs such as BPF and BPAF showed similar or stronger reproductive and developmental toxicity than BPA in *D. magna*. These findings collectively deepen our understanding of the toxicity of BPA analogs and provide empirical evidence for screening safe alternatives to BPA.

## 1. Introduction

Bisphenols (BPs) are a class of compounds used to synthesize materials such as polycarbonate and epoxy resin, which affect the daily lives of people worldwide [1]. Bisphenol A (BPA), an environmental endocrine disruptor, is one of the most widely used compounds [2,3]. Owing to its heavy estrogenic-like effects, its use has been banned or strictly restricted in some nations [4]. In recent years, with the ban or restriction of BPA, the production of BPA analogs with structures similar to those of BPA, including bisphenol S (BPS), bisphenol F (BPF), and bisphenol AF (BPAF), has continued to expand [5,6]. Owing to the rapidly expanding applications of bisphenol analogs, increasing attention has been paid to their toxicity and environmental side-effects.

Several studies have shown that BPs are commonly detected in various environmental media such as air [7], water [8], sediment [8], organisms [9], and human bodies [10,11]. Residual concentrations of BPA and its analogs have been reported to vary from nanograms per liter to milligrams per liter [5,8,12]. Previous studies reported that the highest levels of BPA in Africa were 251 ng/mL, 384.8 ng/mL, and 208.55 ng/mL for water, wastewater, and biological fluids, respectively [13]. Moreover, Wang, et al. [14] found that BPA, BPAF, and BPS were the most predominant analogs in Taihu Lake, China, with residual concentrations ranging from 49.7 to 3480 ng/L.

Generally, the toxicity of substitutes should be weaker than that of the original; however, studies have shown that BPs may have higher toxicity than BPA [14,15]. Several studies have shown that environmentally relevant concentrations of BPA and its analogs may cause abnormalities in mammalian or fish cell viability [16], growth [17], the digestive system [18], the nervous system [19], vascular development [20], and reproduction [21,22].

*D. magna* is a typical model organism used in toxicological studies to evaluate the environmental effects of pollution [23,24,25]. As a major consumer of algae and a primary food source for fish, *D. magna* plays a crucial role in food webs, suggesting that their reproduction may affect the ecosystem [26]. They are easy to handle and have a comparatively short longevity. Moreover, they are known to be quite sensitive to many chemicals [27]. Behavioral disorders in *D. magna* may influence the balance of the aquatic ecosystem. When exposed to environmental pollution, the behavioral responses of *D. magna* directly reflect the occurrence of in vivo sensitivity [28,29]. Liu, et al. [30] found that *D. magna* showed an overcompensatory effect on feeding behavior after short-term exposure to BPA, BPF, and BPS, whereas reproductive behavior did not return to normal levels in long-term exposure tests. Wolstenholme, et al. [31] demonstrated that gestational exposure to low-dose BPA may affect social interactions, particularly in females, but does not affect social preferences in juveniles. However, few studies have focused on highlighting the health concerns regarding the use of BPA alternatives in *D. magna*. 

The aim of our study was to investigate the developmental and reproductive toxicity of *D. magna* exposed to BPA and its analogs, BPS, BPF, and BPAF. We exposed F0 generation *D. magna* to an environmentally relevant concentration of these chemicals (100 μg/L) for 21 d and collected the F1 generation on day 21 (d21). The developmental and reproductive indices, as well as behavioral traits, were recorded on days 7, 14, and 21 (d7, d14, and d21). The expression of genes related to growth and reproduction was detected on d21. Oxidative stress was also investigated. 

## 2. Results

### 2.1. Development and Reproduction of the F0 Generation

No mortality of the F0 generation was observed during the experimental period. The growth, development, and reproduction of the F0 generation are shown in Figure 1. Compared with the control group, the body length of *D. magna* was significantly shorter after exposure to both BPF and BPAF for 7 d (Figure 1b,c). As time progressed, *D. magna* in the BPA group became significantly shorter on 14 d and 21 d. Following BPA, BPF, and BPAF treatment, the heart rates were all significantly reduced at each time point (Figure 1d). Thoracic limb jittering, which can represent movement ability at some levels, exhibited a similar trend to heart rate (Figure 1e). With respect to reproduction, BPF and BPAF treatments may postpone the time to the first brood (Figure 1f). There was also a significant reduction in the number of cumulative offspring in the BPA, BPF, and BPAF groups (Figure 1g). *D. magna* reproduction is usually accompanied by molting. Molting frequency was also reduced in the three groups (Figure 1h). In addition, *D. magna* in the BPS treatment group did not show any significant differences in all of the six parameters.

### 2.2. Development and Reproduction of the F1 Generation

Figure 2 shows the growth, development, and reproduction of the F1 generation. The body length of the offspring was similar to that of the F0 generation, with *D. magna* exposed to BPF and BPAF for 14 or 21 d showing a significant decrease in body length (Figure 2a,b). Offspring of the BPA, BPF, and BPAF maternal treatment groups exhibited a reduced heart rate compared with that of the control group. Over time, this occurrence gradually became more evident (Figure 2c). The trend in thoracic limb tremors was similar to that of the heartbeat, both showing a reduction in the beating rate of the offspring after BPA, BPF, and BPAF treatment at 14 d and 21 d (Figure 2d). Additionally, the reproduction of the F1 generation was influenced by exposure to BPA and its analogs, and this manifested specifically as the deferred first brood time, reduction in cumulative offspring, and decreasing molting frequency (Figure 2e–g).

### 2.3. Behavioral Response

The behavioral response of the F0 generation is shown in Figure 3. As shown in Figure 3a–c, *D. magna* tended to be static in the treatment group, especially in the BPF and BPAF treatment groups, after 21 d of exposure. After 7 d of exposure, there was no significant difference in swimming speed between the control and exposed groups. The swimming speeds of *D. magna* treated with BPS, BPA, BPF, and BPAF were 1%, 9%, 22% (*p* < 0.05), and 36% (*p* < 0.01) of the control group, respectively, after 14 d of exposure and 1%, 20% (*p* < 0.001), 31% (*p* < 0.001), and 33% (*p* < 0.001) after 21 d (Figure 3d).

Figure 4 shows the behavioral response of the F1 generation. The behavioral traits of the F1 generation had a similar tendency to those of the F0 generation. Compared with the control group, the F1 generation in the exposed group moved more sluggishly, especially in the BPF and BPAF groups (Figure 4a–c). The F1 generation differed from the F0 generation with regard to a significant decrease in swimming speed after 7 d of exposure, which increased after 14 d and 21 d (Figure 4d). These results indicated that these chemicals had intergenerational effects on the behavioral response of *D. magna*.

### 2.4. Reflection of Antioxidant Enzymatic Activity

BPS, BPA, BPF, and BPAF may have a greater effect on the F0 generation than the F1 generation. The antioxidant enzymatic activities in almost all the samples collected from the F0 generation (except for the activity of MDA with BPS treatment) were significantly increased (Figure 5a–c), whereas only SOD in the BPA group, MDA in the BPF group, and CAT and MDA in the BPAF group showed significant increases among these indexes (Figure 5d–f).

### 2.5. Expressions of Genes Related to Development and Reproduction

Figure 6 shows that, in the F0 generation, the expression of genes related to development significantly increased after BPAF exposure, whereas *cyp314*, *ecra*, *ecrb,* and *usp* expression significantly increased after BPA exposure. BPF significantly increased the expression of *cyp314* and *ecrb*, whereas BPS significantly increased the expression of *ftz-f1*. The reproduction of *D. magna* was visibly influenced after exposure to BPA and its analogs; *vtg1* and *vtg2* expression improved significantly with BPA, BPF, and BPAF treatment, and *vmo1* expression was inhibited by BPA and BPF treatment. Additionally, in the F1 generation, BPAF significantly increased the expression of *ftz-f1* and *cht* in the offspring and BPF significantly increased the expression of *usp,* whereas BPS inhibited *ecrb* expression.

## 3. Discussion

With the strict restriction of BPA in recent years, BPA analogs have been increasingly used worldwide [5,6]. It has been reported that these chemicals potentially have toxic effects on vertebrates. Previous studies elaborated on the neurotoxicity of BPA and its analogs in zebrafish [15]. Ji, et al. [20] compared and evaluated the vascular toxicity and oxidative stress potency of the BPs in zebrafish and ranked them as follows: BPAF > BPF > BPA > BPS. Moreman, et al. [32] illustrated that BPA and its analogs (BPAF, BPF, and BPS) hindered the development of larval zebrafish, leading to cardiac edema, spinal malformation, and craniofacial deformities. However, few studies have tested the toxicity of BPs in arthropods. Here, we conducted a comprehensive study of the developmental and reproductive toxicity of BPA, BPS, BPF, and BPAF in *D. magna* and assessed their toxic effects. Our results showed that after exposure to BPA and its analogs, both the F0 and F1 generations of *D. magna* exhibited abnormalities in development and reproduction.

Body length can reflect the developmental situation at some levels. Our study showed that, compared with the control group, the body length in the exposed group was reduced, especially in the BPF and BPAF groups, and this inhibition became clearer over time. Heart rate and thoracic limb jittering are linked to feeding behavior or respiratory metabolism, indicating the health and growth of organisms and the stress response to pollutants [33]. Both were significantly decreased in the F0 and F1 generations after BPA, BPF, and BPAF treatment, whereas there was no significant difference in the group exposed to BPS. The thoracic limb is the food-intake organ of *D. magna,* and alterations in the thoracic limb may result in feeding difficulties. It is possible that the depression of heart rate led to a decrease in thoracic limb frequency or that BPAF affected the nervous system [34,35].

Reproductive ability is an important index for detecting the health of the population of *D. magna* and is widely used in contaminant toxicity risk assessment [30]. Although molting frequency is not directly related to spawning ability, some studies have shown a positive correlation between molting and spawning times [36]. We found that the time to the first brood was postponed, whereas the molting frequency was declining in the F0 generation. The cumulative offspring decreased in the F0 generation, which meant that reproduction was influenced by exposure. Moreover, this occurrence did not return to normal levels in juveniles, which indicated that the damage to reproductive capacity in the exposed group may have an intergenerational effect. 

In this study, when exposed to BPA and its analogs, *D. magna* appeared to inhibit locomotor behavior in the F0 generation as well as in the F1 generation. Similar phenomena have also been observed in other animals, exhibiting inhibition of behavior. Gu illustrated that after BPS, BPF, BPA, and BPAF exposure, zebrafish appeared to suffer oxidative stress, larval hypoactivity, and dysregulated neuronal development [15]. Kim, et al. [37] demonstrated that exposure to BPS or other EDs can cause anxiety and reduced social behavior in juvenile mice. In terms of effects during generations, Wolstenholme, et al. [38] found that BPA exposure may influence the social recognition of both parent and descendant rats and has a long-lasting and transgenerational effect even after three generations of BPA exposure, which was consistent with our results. 

It has been reported that BPs enhance reactive oxygen species (ROS) formation and damage the structures of lipids and proteins, which can lead to oxidative damage of cells [39,40]. Previous studies have shown that BPA may induce oxidization system disorders in the liver and nervous system [41,42]. Our results showed that exposure to all four BPs significantly increased the activity of CAT and SOD, and three of the BPs (except BPS) reciprocally increased the levels of MDA, indicating that BPs may induce oxidative stress in *D. magna*.

As mentioned previously, developmental effects and reproductive impairment may have intergenerational effects, which can often be demonstrated using RT-qPCR analysis. The development and reproduction of *D. magna* may be linked to molting, which can be regulated by ecdysteroids (20-hydroxyecdysone, 20-E) and juvenile hormones (JH) [36]. Cytochrome P450s (CYPs) are a superfamily of heme proteins that are vital for drug metabolism, bioactivation, and breakdown of xenobiotics [43,44]. *Cyp314* belongs to this family and is responsible for synthesizing the molting hormone 20-E and converting it to its active form, which can regulate the molting and reproduction cycles [45,46]. In contrast to previous studies [47], our study showed that there is an up-regulation of *cyp314* expression in *D. magna* of F0 generation after BPA, BPF, and BPAF exposure, which may account for the different exposure conditions. When ecdysone 20-E is combined with the protein ecdysone receptor (EcR) and ultraspiracle (USP) to form a nuclear heterodimer complex (EcR/USP-20E), the expression of *ecra*, *ecrb*, and *usp* is altered [45,48]. Moreover, *ftz-f1* may also be expressed after the presentation of 20-E and incentivize the expression of downstream *cht* and other genes [49]. The *cht* gene is involved in cuticle metabolism [47]. The dysregulation of *ecra*, *ecrb*, *usp*, *ftz-f1*, and *cht* in both the F0 and F1 generations may indicate that BPA and its analogs indeed had endocrine effects and deteriorated the molting and metabolism of *D. magna*. Owing to the abnormal fluctuation still occurring in the F1 generation, it can be deduced that the damage may have intergenerational effects in terms of development. 

Exposure to BPA and its analogs led to abnormalities in the gene expression of *D. magna*, which were related to both development and reproduction. As the precursor of vitellin, vitellogenin (*vtg*) is considered an energy source for the development of offspring in oviparous organisms [50,51]. Gene *vom1* prevents the yolk from mixing with the albumen and protecting the egg from bacterial infection [52]. In our study, the expression of *vtg1* and *vtg2* improved significantly with BPA, BPF, and BPAF treatment, whereas the expression of *vmo1* was inhibited by BPA and BPF treatment, which indicated that BP exposure impaired the reproductive system of *D. magna*. Several studies have demonstrated that endocrine disruptors can alter the expression of reproductive genes in organisms, and dysregulation of genes can be transferred during generations [50,53,54].

## 4. Materials and Methods

### 4.1. Experimental Animals

*D. magna* was provided by the Institute of Hydrobiology (Wuhan, China). The culturing method was performed as previously described [55]. Before the experiment, the organisms were incubated for ten generations in deionized water after aeration for 48 h (22 ± 1 °C, pH = 7.50 ± 0.15, salinity 200–230 mg/L). The lighting conditions were a 16 h light/8 h dark cycle under a light intensity of 1000 lx. *D. magna* were fed 2 mL of purebred *Scenedesmus obliquus* (approximately 10^6^ cells mL^−1^ per organism) twice daily. The exposure solutions were changed every 2 d. The non-first larval *D. magna* was selected for toxicity exposure tests based on the OECD211 (Organization for Economic Co-operation and Development Guide standard method) [27].

### 4.2. Chemicals and Test Solutions

BPA (purity: 99%, CAS: 80-05-7), BPF (purity: 99%, CAS: 620-92-8), BPAF (purity: 99%, CAS: 147861-1), and BPS (purity: 99%, CAS: 80-09-1) were purchased from Bailingway Technology Co., Ltd. (Shanghai). Dimethyl sulfoxide (DMSO) was purchased from Sigma-Aldrich. *Chlorella* spp. and *Plagiostellae* spp. were purchased from Wuhan Institute of Hydrology. These four types of BPs were prepared in stock solutions at a concentration of 10^5^ mg/L.

### 4.3. Chronic Toxicity Test

The concentrations were determined to be 100 μg/L according to the environmentally relevant concentration [13], and the stock solution was diluted to the final test solution with deionized water. For the exposure treatment, each group had ten replicates and each replicate had one *D. magna* (F0 generation), which were randomly separated into 100 mL sterile beakers with 80 mL test solution. All the control groups of the other chemicals received 0.01% (*v*/*v*) DMSO. *D. magna* fasted during the test, and other culture conditions were consistent with those described in Section 4.1. The number of new offspring was counted and recorded daily. The offspring (F1 generation) born on days 7, 14, and 21 after maternal exposure (7 d, 14 d, and 21 d) were collected and transferred into clean water [56,57]. Furthermore, molting, survival, and reproduction (both F0 and F1 generations) were recorded at 7 d, 14 d, and 21 d. In addition, behavior tests were performed at these three time points. After 21 d of culture, samples (both F0 and F1 generations) were collected for the following tests.

### 4.4. Heartbeat, Body Length, and Thoracic Limb Beat Frequency Measurement

A *D. magna*, which was able to move (including its antennae) at least every 15 s was considered to be alive [58]. Ten F1 generations were randomly selected from each replicate. F0 and F1 generations were observed by a Research Stereo Microscope (SMZ25, Nikon, Japan). The length from the top of the helmet to the base of the tail (excluding the tail spur) was measured as the body length of *D. magna*. The number of heart beats and the number of thoracic limb beats that survived for 1 min were recorded by manual counting as heart beats (one per minute) and thoracic limb shakes (one per minute).

### 4.5. Behavioral Tests

Behavioral tests were conducted using a DanioVision observation chamber (Noldus, Wageningen, Netherlands). According to the method of a previous study [23], 10 *D. magna* F0 or F1 generations were selected from each exposure treatment. A 48-well plate was used for detection, with one *D. magna* in each well filled with 1 mL of exposure solution. The test program was set for 35 min, including 5 min of dark acclimation and three 10-min light-and-dark cycles, which consisted of 5 min of light and 5 min of darkness. Movement traits, heat maps, and swimming speed were analyzed using EthoVision XT 15 video tracking software (Noldus, Wageningen, The Netherlands).

### 4.6. Real-Time Quantitative Reverse Transcription-Polymerase Chain Reaction (RT-qPCR)

After 21 d of exposure, ten *D. magna* from each exposure treatment were mixed for further study of gene transcription using TRIzol reagent (Thermo Fisher, Waltham, MA, USA), according to the manufacturer’s instructions. RNA concentration was measured using a NanoDrop2000 (Thermo Fisher), and cDNA was synthesized using a PrimeScript^®^ RT reagent kit (TaKaRa, Shiga, Japan). The expression of the following genes was measured: *cytochrome p450 314* (*cyp314*), *ecdysone receptor a* (*ecra*), *ecdysone receptor b* (*ecrb*), *ultraspiracle* (*usp*), *fushi tarazu factor-1* (*ftz-f1*), *chitinase* (*cht*) (which are related to development) and *vitellogenin 1* (*vtg1*), *vitellogenin 2* (*vtg2*), *vitelline outer layer membrane protein 1* (*vmo1*) (which are related to reproduction). The primers of these genes are shown in Appendix A.

### 4.7. Determination of Antioxidant Enzymatic Activity

The lysate was homogenized using an ultrasonic crushing instrument and the protein concentration was determined using a protein detection kit (Biyuntian, Shanghai, China). Superoxide dismutase (SOD; Biyuntian) and catalase (CAT; Biyuntian) (MDA; Biyuntian) detection kits were used to determine the enzymatic activities of the different treatment groups. 

### 4.8. Statistical Analysis

The data were analysed with one-way analysis of variance (ANOVA) using GraphPad Prism 8.0.1 software. The results are expressed as the mean value ± standard error of mean (SEM). Asterisks indicate significant differences between exposure and control, * indicates a significant difference (* *p* < 0.05), ** indicates a very significant difference (** *p* < 0.01), and *** indicates a highly significant difference (*** *p* < 0.001).

## 5. Conclusions

BPA has been strictly restricted because of its strong endocrine effects, and analogs have been invented for broad applications. However, the toxicity of these substitutes has not been fully evaluated. In the present study, we found that BPA alternatives induce similar toxic and estrogenic effects to BPA, and the toxicity of its three BPA substitutes (BPS, BPF, and BPAF) on the growth and reproduction of *D. magna* may be not weaker than those of BPA, which manifest as growth inhibition, oxidative stress, and altered gene expression. Considering all aspects, the toxicity of these four chemicals in *Daphnia magna* should be in the order BPAF > BPF ≈ BPA > BPS, which is consistent with the findings of previous studies conducted on zebrafish. Given that BPAF and BPF have toxic effects comparable to those of BPA, BPAF and BPF should be carefully considered as alternatives to BPA. In addition, we should be more concerned about the long-term low-dose effects of such alternatives at environmental concentrations. We can step up publicity to establish the perceptions of BPs’ impacts, adjust lifestyle habits to reduce daily exposure, and establish corresponding laws and regulations to cut off the emission of BPs [4].

## Figures and Tables

**Figure 1 ijms-23-14561-f001:**
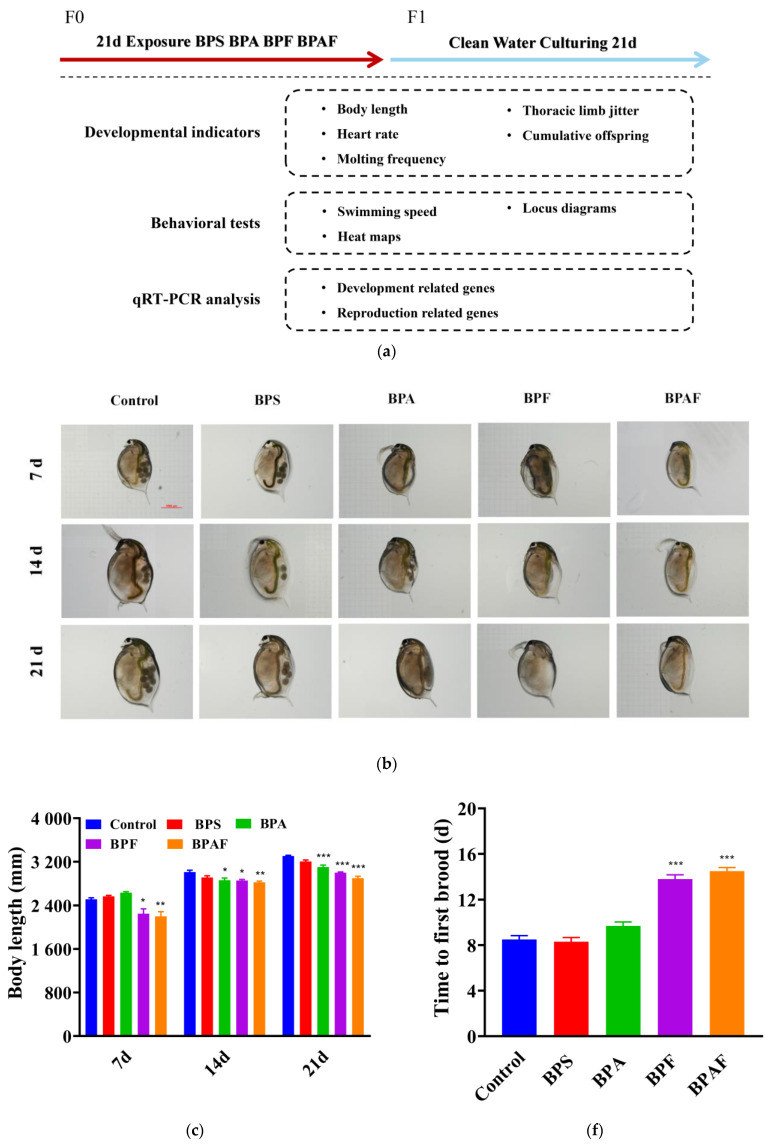
A schematic of the experimental design used (**a**). Development and reproduction of the F0 gener-ation induced by BPs: *D. magna* with microscopic observation (**b**), body length (**c**), heart rate (**d**), thoracic limb jittering (**e**), time to first brood (**f**), cumulative offspring (**g**), and molting frequency (**h**). Asterisks indicate significant differences between exposed and control groups (* *p* < 0.05, ** *p* < 0.01, *** *p* < 0.001).

**Figure 2 ijms-23-14561-f002:**
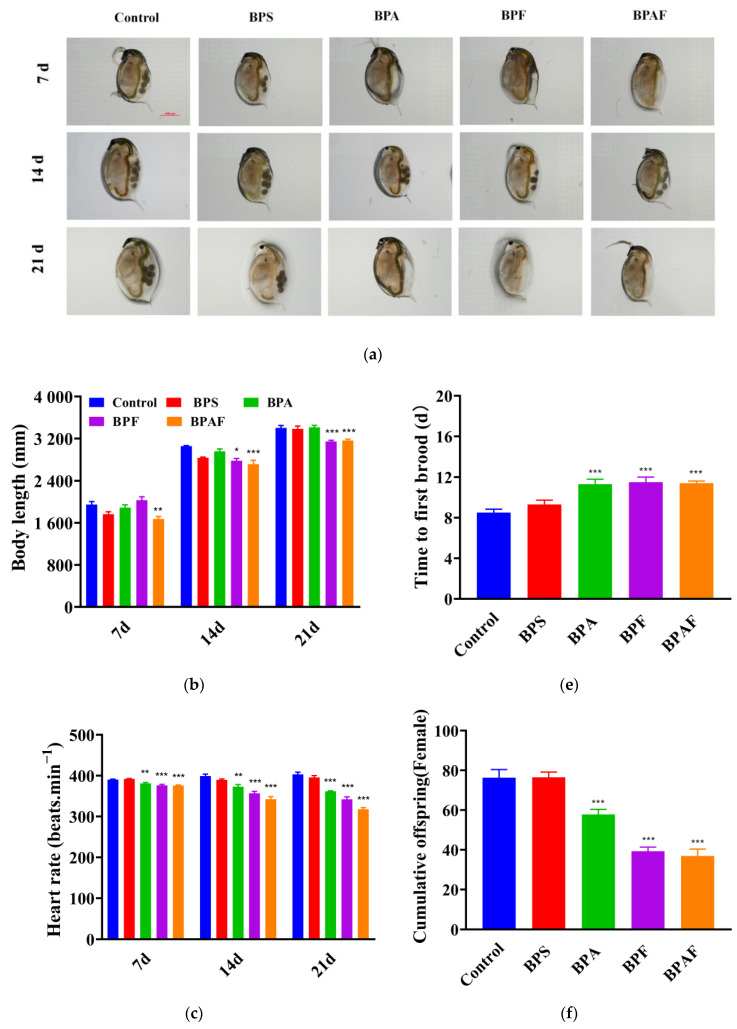
Development and reproduction of the F1 generation induced by BPs. *D. magna* with microscopic observation (**a**), body length (**b**), heart rate (**c**), thoracic limb jittering (**d**), time to first brood (**e**), cumulative offspring (**f**), and molting frequency (**g**). Asterisks indicate significant differences be-tween exposed and control groups (* *p* < 0.05, ** *p* < 0.01, *** *p* < 0.001).

**Figure 3 ijms-23-14561-f003:**
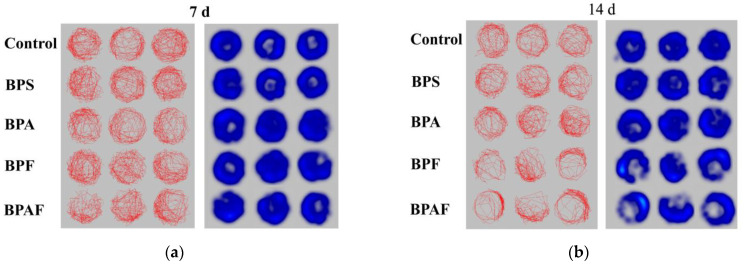
Behavioral response of the F0 generation. Locus diagrams and heat maps after 7 (**a**), 14 (**b**), and 21 (**c**) d exposure. Swimming speed (**d**). Asterisks indicate significant differences between the exposed and control groups (** *p* < 0.01, *** *p* < 0.001).

**Figure 4 ijms-23-14561-f004:**
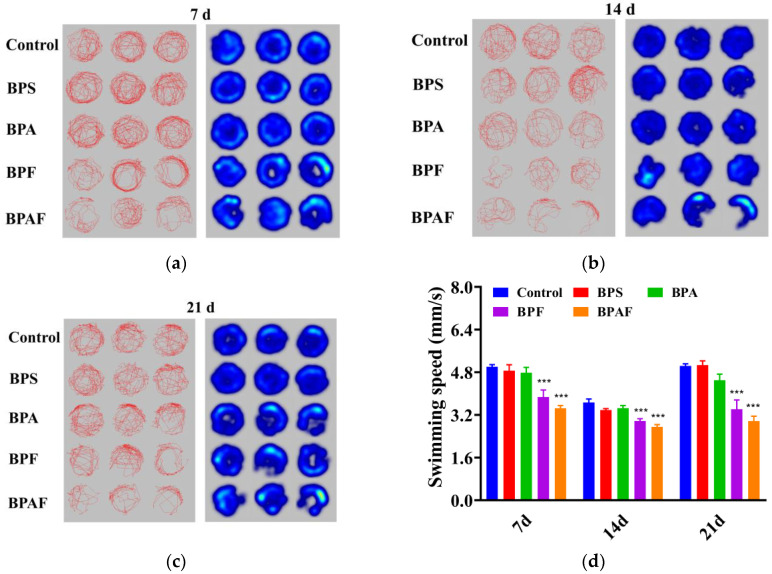
Behavioral response of the F1 generation. Locus diagrams and heat maps after 7 (**a**), 14 (**b**), and 21 (**c**) d exposure. Swimming speed (**d**). Asterisks indicate significant differences between the exposed and control groups (*** *p* < 0.001).

**Figure 5 ijms-23-14561-f005:**
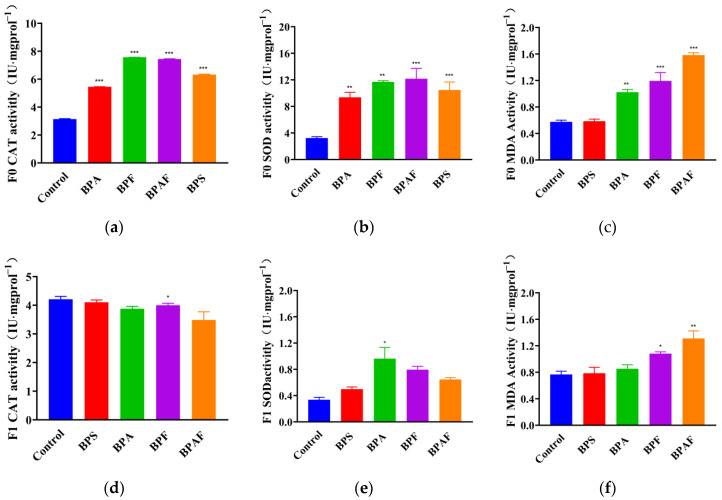
Reflection of antioxidant enzymatic activity of the F0 generation (**a**–**c**) and the F1 generation (**d**–**f**). Enzymatic activity of CAT (**a**,**b**), SOD (**b**,**e**), and MDA (**c**,**f**). Asterisks indicate significant differ-ences between the exposed and control groups (* *p* < 0.05, ** *p* < 0.01, *** *p* < 0.001).

**Figure 6 ijms-23-14561-f006:**
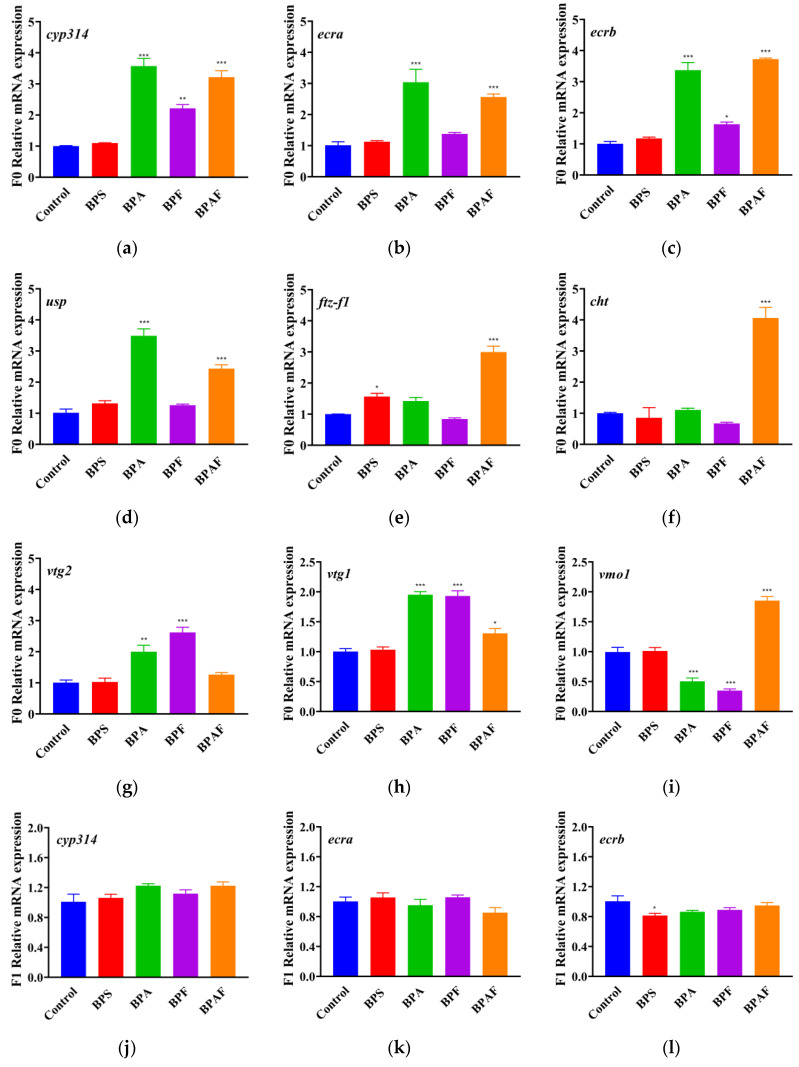
Expression of genes related to development (**a**–**f**,**j**–**o**) and reproduction (**g**–**i**) in the F0 generation (**a**–**i**) and F1 generation (**j**–**o**). Asterisks indicate significant differences between the exposed and control groups (* *p* < 0.05, ** *p* < 0.01, *** *p* < 0.001).

## Data Availability

Not applicable.

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
