# Peer review of "Developmental and Reproductive Impacts of Four Bisphenols in *Daphnia magna"

_ijms, 2022, doi:10.3390/ijms232314561_

Round 1

Reviewer 1 Report

The authors of the manuscript entitled “The developmental and reproductive toxicology of Daphnia 2 magna after exposure to four bisphenols” investigated the developmental and reproductive impact of BPA and its analogs, BPS, BPF, and BPAF in D. magna. The authors found that BPA alternatives induce similar toxic and estrogenic effects as BPA, and the toxicity of its three BPA substitutes (BPS, BPF, BPAF) on the growth and reproduction of D. magna may be not weaker than that of BPAThe study is novel and has the potential to be accepted for publication provided the authors can address all major issues listed below.

1.               The title is confusing, the topic should be rephrased, e.g., developmental and reproductive impacts of four bisphenols in Daphnia magna

2.               Lines 16 and 36. BPA is not strictly restricted globally, only a few nations completely banned it (PMID: 34004580)

3.               The contents of the manuscript in several places do not align, for example, the authors claimed that BPA is restricted in line 36 but also claimed its expanding application in line 40

4.               Lines 61 and 63, Liu, and Wolstenholme should be appropriately referenced

5.               Lines 69 to 76 should be deleted from the introduction; this section can be taken to the result section.

6.               The author should include a diagram that concisely described the measurements

7.               The procedure for the measurement of the heartbeat is not detailed enough. The author should describe this procedure in detail.

8.               The quality of the figures, especially the bar graphs is very poor. Kindly improve

9.               Figure 2C (14d), Is the author sure that BPS is statistically different from BPA?

10.            Line 303, please rephrase the sentence and other incorrect sentences in the entire manuscript.

11.            Line 315, are bisphenols drugs? Kindly change it.

12.            The author should indicate that the order of the toxicity of the chemical as listed in line 316 is in D. magna. This order could be different in other species.

13.            While several experts and toxicology experts have been advocating for the ban of these chemicals, while is author dragging the public back by encouraging the cautious use of these chemicals. Also, the cautious use that author mentioned is not specific. 

Author Response

Point 1: The title is confusing, the topic should be rephrased, e.g., developmental and reproductive impacts of four bisphenols in Daphnia magna.

 Response 1: Thank you for the suggestion. We revised the title of this MS to ‘Developmental and reproductive impacts of four bisphenols in Daphnia magna’, following your helpful suggestion.

Point 2: Lines 16 and 36. BPA is not strictly restricted globally, only a few nations completely banned it (PMID: 34004580).

Response 2: Thanks for the suggestion. The description has been revised to ‘Considering its adverse effects, BPA has been banned or strictly restricted in some nations, and many analogs have been introduced to the market.’, as shown on Page 1, Lines 16-17, and ‘Owing to its heavy estrogenic-like effects, its use has been banned or strictly restricted in some nations [1].’, as shown on Page 1, Lines 36-37.

Point 3: The contents of the manuscript in several places do not align, for example, the authors claimed that BPA is restricted in line 36 but also claimed its expanding application in line 40.

Response 3: Thanks for the comment. We revised this statement to ‘Owing to the rapidly expanding applications of bisphenol analogs, increasing attention has been paid to their toxicity and environmental side effects.’, as shown on Page 1, Lines 40-41.

Point 4: Lines 61 and 63, Liu, and Wolstenholme should be appropriately referenced.

Response 4: Thanks for the comment. The description has been revised to ‘Liu, et al. [2] found that D. magna showed an overcompensatory effect on feeding behavior after short-term exposure to BPA, BPF, and BPS, whereas reproductive behavior did not return to normal levels in long-term exposure tests. Wolstenholme, et al. [3] demonstrated that gestational exposure to low-dose BPA may affect social interactions, particularly in females, but does not affect social preferences in juveniles.’ (Page 2, Lines 62-67), and all descriptions in the following manuscript have been changed to have a consistent format.

Point 5: Lines 69 to 76 should be deleted from the introduction; this section can be taken to the result section.

Response 5: Thanks for the comment. We deleted this section ‘Our results showed that after exposure to BPA and its analogs, both the F0 and F1 generations of D. magna exhibited abnormalities in development and reproduction, whereas the toxicity of BPAF was the highest, followed by BPF and BPA, and BPS was the lowest.’ and move it to the result on Page 10, Lines 168-170.

Point 6: The author should include a diagram that concisely described the measurements.

Response 6: Thanks for the suggestion. We have added a supply that concisely describes the measurement results. The schematic is like the following.

(see attached pdf)

Point 7: The procedure for the measurement of the heartbeat is not detailed enough. The author should describe this procedure in detail.

Response 7: Thanks for the suggestion. The description has been revised to ‘Ten F1 generations were randomly selected from each replicate. F0 and F1 generations were observed by a Research Stereo Microscope (SMZ25, Nikon, Japan). The length from the top of the helmet to the base of the tail (excluding the tail spur) was measured as the body length of D. magna. The number of heart beats and the number of thoracic limb beats that survived for 1 min were recorded by manual counting as heart beats (one per minute) and thoracic limb shakes (one per minute).’ (Page 12, Lines 272-277).

Point 8: The quality of the figures, especially the bar graphs is very poor. Kindly improve.

Response 8: Thanks for the suggestion. We have revised all the figures in the article.

Point 9: Figure 2C (14d), Is the author sure that BPS is statistically different from BPA?

 Response 9: Thanks for the comment. We did have a copy error in the data here, which has been corrected. All the data in this article has been rechecked and no other errors have been found.

Point 10: Line 303, please rephrase the sentence and other incorrect sentences in the entire manuscript.

 Response 10: Thanks for the comment. We revised this statement to ‘The statistics were analyzed with one-way analysis of variance (ANOVA) using GraphPad Prism 8.0.1 software. The results are expressed as the mean value ± standard error of mean (SEM).’, as shown on Page 12, Lines 304-306. All descriptions in the entire manuscript have been changed to have a consistent format.

Point 11: Line 315, are bisphenols drugs? Kindly change it.

Point 12: The author should indicate that the order of the toxicity of the chemical as listed in line 316 is in D. magna. This order could be different in other species.

 Response 11&12: Thanks for the comment. We revised the description to ‘Considering all aspects, the toxicity of these four chemicals in Daphnia magna should be in the order BPAF > BPF ≈ BPA > BPS, which is consistent with the findings of previous studies conducted on zebrafish.’ as shown on Page 13, Lines 316-318.

Point 13: While several experts and toxicology experts have been advocating for the ban of these chemicals, while is author dragging the public back by encouraging the cautious use of these chemicals. Also, the cautious use that author mentioned is not specific.

 Response 13: Thanks for the comment. We revised the description to ‘Given that BPAF and BPF have toxic effects comparable to those of BPA, BPAF and BPF should be carefully considered as alternatives to BPA, in addition to being more concerned about the long-term low-dose effects of such alternatives at environmental concentrations. We can step up in publicity to establish the perceptions of BPs impacts, adjust the adoption of lifestyle habits to reduce daily exposure, and establish corresponding laws and regulations to cut off the emission [1].’(Page 13, Lines 318-323).

  1. Rahman, M. S.; Adegoke, E. O.; Pang, M. G., Drivers of owning more BPA. J Hazard Mater 2021, 417, 126076.
  2. Liu, Y.; Yan, Z.; Zhang, L.; Deng, Z.; Yuan, J.; Zhang, S.; Chen, J.; Guo, R., Food up-take and reproduction performance of Daphnia magna under the exposure of Bisphenols. Ecotoxicol Environ Saf 2019, 170, 47-54.
  3. Wolstenholme, J. T.; Taylor, J. A.; Shetty, S. R.; Edwards, M.; Connelly, J. J.; Rissman, E. F., Gestational exposure to low dose bisphenol A alters social behavior in juvenile mice. PLoS One 2011, 6, (9), e25448.

Reviewer 2 Report

* I recommend adding a schematic of the experimental design used.

* Why did you use animals born at 7, 14, 21 days?

* Could you please provide more information on the primers used? (RT-qPCR)

* Do other international groups outside your country use these methods?

* Are there similar results in other parts of the world with which to compare your results?

* Why do you only use one dose of bisphenols? Why always the same one? You provide 2 references, but you should explain it better in the text.

* It would have been interesting to test the effects of mixing various bisphenols. Is there any reason why they didn't?

* What advantages does this animal model have over the use of zebrafish?

Author Response

Point 1: I recommend adding a schematic of the experimental design used.

Response 1: Thanks for the comment. We have already added the schematic in the article. The schematic is like the following.(See attached pdf)

Point 2: Why did you use animals born at 7, 14, 21 days?

Point 4: Do other international groups outside your country use these methods?

Response 2&4: Thank you for the comment. We conducted the experiment following OECD guidelines [1] and selected these three time points according to previous studies [2, 3]. These three time points are important for the growth and development of Daphnia magna. Moreover, we added the references to the article, as shown as shown on Page 11-12, Lines 263-266.

Point 3: Could you please provide more information on the primers used? (RT-qPCR).

Response 3: Thanks for the suggestion. We have added a supplement that includes the information on the primers used.

Point 5: Are there similar results in other parts of the world with which to compare your results?

Response 5: Yes. Previous studies elaborated the similar results in zebrafish. For example, Ji, et al. [4] found that the vascular toxicity and oxidative stress potency of the BPs were compared and evaluated, ranking as follows: BPAF > BPF > BPA > BPS in zebrafish. Moreman, et al. [5] illustrated that BPA and its analogs (BPAF, BPF, and BPS) hindered the development of larval zebrafish, leading to cardiac edema, spinal malformation, and craniofacial deformities. We have added these references in paper, as shown on Page 9-10, Lines 161-165.

Point 6: Why do you only use one dose of bisphenols? Why always the same one? You provide 2 references, but you should explain it better in the text.

Response 6: Thank you for the comment. The main purpose of our study was to compare the toxicity of the four BPs at reference ambient concentrations (251 ng/mL in water and 384.8 ng/mL in waste water) [6]. Choosing the same concentration was more conducive to comparing toxicity. The results can be more intuitive to illustrate the toxicity at the same concentration. We reconfirmed the reference we used and revised it as shown on Page 11, Lines 257-259.

Point 7: It would have been interesting to test the effects of mixing various bisphenols. Is there any reason why they didn't?

Response 7: Thanks for your comment. Our aim is to compare the developmental and reproductive impacts of each four substances. Your comments have inspired us. It is an interesting topic to study the effects of mixing various bisphenols. We could examine the effects of mixed exposures in Daphnia magna in the following studies.

Point 8: What advantages does this animal model have over the use of zebrafish?

Response 8: D. magna is a typical model organism to evaluate the environmental effects of pollution. They are major consumers of algae and primary food source for fish which plays a crucial role in food webs [7]. Moreover, they are known to be quite sensitive to many chemicals. Compared to zebrafish, they are easier to handle and have a comparatively short longevity [1]. We have already added these features in the paper as shown Page 2, Lines 55-59.

  1. OECD, Test No. 211: Daphnia magna Reproduction Test. 2012.
  2. Blewett, T. A.; Delompre, P. L.; He, Y.; Folkerts, E. J.; Flynn, S. L.; Alessi, D. S.; Goss, G. G., Sublethal and Reproductive Effects of Acute and Chronic Exposure to Flowback and Produced Water from Hydraulic Fracturing on the Water Flea Daphnia magna. Environ Sci Technol 2017, 51, (5), 3032-3039.
  3. Bao, S.; Pan, B.; Wang, L.; Cheng, Z.; Liu, X.; Zhou, Z.; Nie, X., Adverse effects in Daphnia magna exposed to e-waste leachate: Assessment based on life trait changes and responses of detoxification-related genes. Environ Res 2020, 188, 109821.
  4. Ji, G.; Gu, J.; Guo, M.; Zhou, L.; Wang, Z.; Shi, L.; Gu, A., A systematic comparison of the developmental vascular toxicity of bisphenol A and its alternatives in vivo and in vitro. Chemosphere 2022, 291, (Pt 2), 132936.
  5. Moreman, J.; Lee, O.; Trznadel, M.; David, A.; Kudoh, T.; Tyler, C. R., Acute Toxicity, Teratogenic, and Estrogenic Effects of Bisphenol A and Its Alternative Replacements Bisphenol S, Bisphenol F, and Bisphenol AF in Zebrafish Embryo-Larvae. Environ. Sci. Technol. 2017, 51, (21), 12796-12805.
  6. Rotimi, O. A.; Olawole, T. D.; De Campos, O. C.; Adelani, I. B.; Rotimi, S. O., Bisphenol A in Africa: A review of environmental and biological levels. Sci Total Environ 2021, 764, 142854.
  7. Ten Berge, W. F., Breeding Daphnia magna. Hydrobiologia 1978, 59, (2), 121-123.

Round 2

Reviewer 1 Report

The authors have addressed the major concerns of the reviewer, however, the author should change the   statement "the statistics were analysed" to "the data were analysed". The manuscript could be accepted upon correction of this minor correction

Author Response

Point 1: The authors have addressed the major concerns of the reviewer, however, the author should change the statement "the statistics were analysed" to "the data were analysed". The manuscript could be accepted upon correction of this minor correction.

Response 1: Thank you for the suggestion. We revised the statement to ‘The data were analysed with one-way analysis of variance (ANOVA) using GraphPad Prism 8.0.1 software.’, as shown on Page 12, Lines 304-305.